# Autonomic Dysfunction during Acute SARS-CoV-2 Infection: A Systematic Review

**DOI:** 10.3390/jcm11133883

**Published:** 2022-07-04

**Authors:** Irene Scala, Pier Andrea Rizzo, Simone Bellavia, Valerio Brunetti, Francesca Colò, Aldobrando Broccolini, Giacomo Della Marca, Paolo Calabresi, Marco Luigetti, Giovanni Frisullo

**Affiliations:** 1School of Medicine and Surgery, Catholic University of Sacred Heart, Largo Francesco Vito, 1, 00168 Rome, Italy; irene.scala92@gmail.com (I.S.); pierandrea.rizzo01@gmail.com (P.A.R.); bellavia.sim@gmail.com (S.B.); colofra94@gmail.com (F.C.); aldobrando.broccolini@policlinicogemelli.it (A.B.); giacomo.dellamarca@policlinicogemelli.it (G.D.M.); paolo.calabresi@policlinicogemelli.it (P.C.); 2Dipartimento di Scienze dell’Invecchiamento, Neurologiche, Ortopediche e Della Testa-Collo, Fondazione Policlinico Universitario Agostino Gemelli IRCCS, 00168 Rome, Italy; v.brunetti2509@gmail.com (V.B.); giovanni.frisullo@policlinicogemelli.it (G.F.)

**Keywords:** COVID-19, SARS-CoV-2, dysautonomia, autonomic nervous system, autonomic dysfunction, heart rate variability, outcome, prognosis

## Abstract

Although autonomic dysfunction (AD) after the recovery from Coronavirus disease 2019 (COVID-19) has been thoroughly described, few data are available regarding the involvement of the autonomic nervous system (ANS) during the acute phase of SARS-CoV-2 infection. The primary aim of this review was to summarize current knowledge regarding the AD occurring during acute COVID-19. Secondarily, we aimed to clarify the prognostic value of ANS involvement and the role of autonomic parameters in predicting SARS-CoV-2 infection. According to the PRISMA guidelines, we performed a systematic review across Scopus and PubMed databases, resulting in 1585 records. The records check and the analysis of included reports’ references allowed us to include 22 articles. The studies were widely heterogeneous for study population, dysautonomia assessment, and COVID-19 severity. Heart rate variability was the tool most frequently chosen to analyze autonomic parameters, followed by automated pupillometry. Most studies found ANS involvement during acute COVID-19, and AD was often related to a worse outcome. Further studies are needed to clarify the role of autonomic parameters in predicting SARS-CoV-2 infection. The evidence emerging from this review suggests that a complex autonomic nervous system imbalance is a prominent feature of acute COVID-19, often leading to a poor prognosis.

## 1. Introduction

The severe acute respiratory syndrome coronavirus 2 (SARS-CoV-2), a novel coronavirus, is the etiological agent of Coronavirus disease 2019 (COVID-19), a predominantly respiratory disease, which spread worldwide over a few months [1]. The spectrum of COVID-19 severity is highly variable, ranging from asymptomatic carriers to severe acute respiratory distress syndrome (ARDS) leading to death [2,3].

Despite the SARS-CoV-2 tropism for the respiratory system, strong evidence pointed to the ability of the virus to induce multiorgan damage [4]. In this context, a neurological involvement was reported in more than one-third of acute COVID-19 patients [5], characterized by a wide range of symptoms, such as smell and taste loss, stroke, encephalopathies, peripheral nervous system disturbances, and, especially after the recovery from the acute SARS-CoV-2 infection, symptoms suggestive of an autonomic dysfunction (AD) [6,7,8,9].

It is not the first time that autonomic nervous system (ANS) disturbances have been associated with an acute viral disease. Evidence of autonomic involvement has already been reported during systemic infections caused by viruses belonging to different families, such as Retroviridae, Herpesviridae, Picornaviridae, Rhabdoviridae, Flaviviridae, Orthomixoviridae, and Pneumoviridae [10,11,12,13]. Focusing on the Coronaviridae family and, in particular, the Coronaviruses species, which share great genetic similarities with SARS-CoV-2 [14], the occurrence of autonomic symptoms has been described in SARS-CoV and MERS-CoV infections, mostly presenting as chronic fatigue following the acute phase of the disease [15,16,17].

Orthostatic cerebral hypoperfusion syndrome [18], small fiber neuropathy [18,19,20], orthostatic hypotension and intolerance [21,22], postural orthostatic tachycardia syndrome (POTS) [19,23], altered pupillary reactivity [24], and heart rate variability (HRV) [25] are the most common symptoms that may last for several weeks or months after the infection recovery in approximately 33–87% of patients [26]. This condition is known as post-COVID-19 syndrome or “long COVID”.

Based on individual studies, it is difficult to determine whether the ANS involvement is an exclusive and specific feature of the long-COVID period or whether dysautonomic symptoms during the acute phase of the disease are overtaken by respiratory or other prevailing symptoms. Indeed, few data are currently available regarding the occurrence of autonomic disturbances during the acute SARS-CoV-2 infection and their possible role in prognosis and early diagnosis of the disease. 

The primary aim of this review is to clarify the features of the ANS involvement during the acute SARS-CoV-2 infection. Secondarily, this review aims to evaluate the prognostic role of autonomic involvement occurring during the acute phase of the disease and to summarize the available evidence regarding the role of autonomic parameters in predicting the SARS-CoV-2 infection.

## 2. Materials and Methods

### 2.1. Eligibility Criteria

This systematic review was conducted according to the latest Preferred Reporting Items for Systematic reviews and Meta-Analyses (PRISMA) recommendations [27]. The eligibility of the reports was defined based on the PICOS criteria, as shown in Table 1.

### 2.2. Search Strategy

The latest research was carried out by three authors (I.S., S.B., and G.F.) on 23 April 2022 using Scopus and PubMed databases. The following string of words was employed to search the pertinent reports: (((COVID-19) OR (COVID 19) OR (COVID) OR (SARS CoV 2) OR (SARS-CoV-2) OR (coronavirus) OR (2019 novel coronavirus)) AND ((dysautonomia) OR (autonomic dysfunction) OR (autonomic) OR (autonomic nervous system) OR (orthostatic hypotension) OR (orthostatic intolerance) OR (exercise intolerance) OR (postural orthostatic tachycardia syndrome) OR (tilt test) OR (sinus tachycardia) OR (pupillometry) OR (Sudoscan) OR (sudomotor dysfunction) OR (heart rate variability) OR (HRV) OR (Valsalva maneuver))). 

Additionally, the reference lists of the selected articles and of pertinent published reviews investigating the occurrence of neurological or dysautonomic symptoms in the context of acute SARS-CoV-2 infection were screened in order to find other eligible records.

### 2.3. Selection Process

After deleting duplicates, three authors (I.S., V.B., and G.F.) independently screened the records by title and abstract. Subsequently, the full text of all the retrieved reports was read and evaluated independently by three authors (I.S., G.F., S.B.) in order to assess the eligibility of the manuscripts. A report was considered eligible for this systematic review when at least two out of three authors agreed on its eligibility.

### 2.4. Data Extraction

Three authors (I.S., S.B., and P.A.R.) retrieved the data from the eligible reports and reported the variables of interest in an Excel file. The following data were extracted: first author and year of publication, design of the study, sample size of the study population (i.e., total number of COVID-19 population, controls, and population subgroups where applicable), demographic features of study subjects (i.e., age and sex), severity of COVID-19 infection, tests employed for COVID-19 diagnoses, tools employed for autonomic dysfunction analysis, study endpoints, and main findings of the study. Any doubtful situation was solved by referring to the other two authors (G.F. and V.B.). The severity of COVID-19 was defined based on the setting in which infected patients were enrolled and the need for mechanical ventilation. In particular, the disease was considered mild when the enrolled subjects were in home quarantine, moderate when the disease required hospitalization in a regular ward or in a sub-intensive care unit, and severe in case of admission to an intensive care unit (ICU) and/or need for mechanical ventilation.

I.S. contacted the corresponding authors of the pertinent studies for which extensive data regarding the autonomic assessment and the prevalence of COVID-19 were missing. Additionally, when not available in the full text, the corresponding authors were asked to provide demographic features of the study sample and any other useful information.

### 2.5. Quality Assessment

Quality assessment of the included studies was performed by means of the “Study Quality Assessment Tool” issued by the National Heart, Lung, and Blood Institute within the National Institute of Health for cohort and cross-sectional studies, case–control studies, and pre–post studies [28]. Two authors (S.B. and P.A.R.) evaluated all the reports independently, and any doubtful situation was solved by referring to the third author (I.S.).

## 3. Results

### 3.1. Study Selection

The online search returned 843 records from PubMed and 1277 from Scopus databases. After removing duplicates, 1585 results remained and were evaluated for title and abstract check. Therefore, 65 reports were considered in the full-text review stage, and 20 of those were included in the systematic review [29,30,31,32,33,34,35,36,37,38,39,40,41,42,43,44,45,46,47,48], while 45 reports were excluded: 2 papers written in Russian [49,50], 5 conference papers [51,52,53,54,55], 3 comments [56,57,58], 4 case reports or case series [59,60,61,62], 2 literature reviews [63,64], 1 study protocol [65], 12 studies that did not perform a formal evaluation of autonomic functions [66,67,68,69,70,71,72,73,74,75,76,77], 13 reports that did not include acute COVID-19 patients [78,79,80,81,82,83,84,85,86,87,88,89,90], and 3 studies in which extensive data regarding the autonomic assessment of the population of interest were lacking [91,92,93].

Additional two studies were found by hand-searches and a cross-search of the references [94,95]. 

Finally, 22 studies were included in this systematic review [29,30,31,32,33,34,35,36,37,38,39,40,41,42,43,44,45,46,47,48,94,95].

Details regarding the selection process are available in the PRISMA diagram (Figure 1).

### 3.2. Study Characteristics

All the studies included in this review were observational: five with a cross-sectional design [31,36,40,46,47]; nine retrospective studies, including four case–control studies [37,41,44,94], two cohort studies [30,43], and three pre–post studies [33,35,95]; eight prospective studies that included seven cohort studies [29,32,34,38,39,42,45] and one pre–post study [48].

The number of COVID-19 patients ranged from 14 [29,95] to 42,752 [39], and in most cases, the diagnosis of SARS-CoV-2 infection was performed through a real-time polymerase chain reaction (RT-PCR) test [29,30,31,32,36,37,38,39,41,42,43,44,45,47,48,94]. In a minority of reports, COVID-19 diagnosis was self-reported by study participants [33,34,95], and in one study, both the RT-PCR test and/or a rise in anti-SARS-CoV-2 IgG levels were considered as diagnostic for COVID-19 [39]. Finally, three studies did not specify the type of test employed for the diagnoses of SARS-CoV-2 infection [35,40,46].

The age of COVID-19 patients was very heterogenous, ranging from a median age of 34 (age range 1–102) years [39] to a mean age of 78.6 ± 11.4 years [35]. However, age range and gender prevalence were hardly definable because of the great variability in data reporting among the included reports and the lack of demographics in two studies [38,94]. 

Concerning the severity of the acute SARS-CoV-2 infection, 16 studies included hospitalized patients [29,30,31,32,35,36,37,38,41,42,43,44,45,47,48,94]. According to the criteria listed above, in five reports, COVID-19 could be defined as severe [29,30,32,37,47], in four articles as moderate [31,36,41,48], while in six studies, patients affected by both severe and moderate forms of the disease were included [35,38,42,43,44,45], and in one study, the severity of the disease was not specified [94]. In two articles, both hospitalized patients and subjects in home quarantine were enrolled [39,40]. In one study, the setting of the enrollment was not specified, but the authors stated that all the SARS-CoV-2-infected patients did not need oxygen support [46]. Finally, two studies analyzed the data reported by all [33] or some [95] of the participants of the Weltory study [96], a worldwide database in which patients consented to share data obtained by their own wearable devices in order to fight against the SARS-CoV-2 pandemic. Although COVID-19 severity, as self-reported by study participants, ranged from asymptomatic to extremely severe forms, specific data regarding the setting and the effective treatments needed are not available. Similarly, Hirten et al. [34] included healthcare workers of the Mount Sinai Health System, reporting data obtained from their wearable devices, but clear information regarding the severity of COVID-19 were missing.

HRV was the most used tool to evaluate the ANS functions [29,32,33,34,35,36,37,38,40,42,43,45,46,94,95], although its recording method varied widely among the studies, ranging from the use of wearable devices, short-duration electrocardiograms (EKGs), and 24 h Holter EKGs or continuous bedside monitoring. Automated pupillometry was chosen by three authors [30,47,48], two studies carried out a composite evaluation of ANS [31,41], and in another two, AD was defined based on the retrospective analysis of patients’ medical records [39,44].

All data extracted from the included reports are summarized in Table 2, Table 3 and Table 4.

### 3.3. Quality Assessment

According to the “Study Quality Assessment Tool” issued by the National Heart, Lung, and Blood Institute [28], three studies were classified as having a “good quality” [30,38,42], eight studies were considered to be of “fair quality” [29,33,36,37,41,43,45,48], and the other eleven were classified as “poor quality” studies [31,32,34,35,39,40,44,46,47,94,95]. 

Blinding was the most frequent risk of bias encountered. Only one study (4.5%) reported in an explicit way that the outcome assessors were blinded to the exposure status of participants [38]. Secondarily, sample size justification was reported in only two studies (9.1%) [29,38]. Statistical adjustment of possible confounding variables was performed in only 5 out of 17 cohort and case–control studies (29.4%) [30,36,38,40,42]. On the other hand, only in three studies (13.6%) was the research question not clearly defined [35,39,40].

A graphic summary of the quality assessment is reported in Figure 2. The extended results of the quality assessment are represented in the Appendix A.

### 3.4. Outcomes

Ten studies focused on the characterization of autonomic involvement during acute SARS-CoV-2 infection as a primary endpoint [31,34,35,36,40,41,46,47,48,94]. In seven articles, the primary aim was to evaluate the impact of AD on COVID-19 prognosis [29,32,37,38,42,43,45], while the analysis of the role of autonomic parameters in predicting SARS-CoV-2 infection was the main objective of two studies [33,95]. Finally, three articles did not focus on autonomic alterations, but the data of interest for this systematic review were extrapolated from the text [39,44] or were provided by the corresponding authors of the manuscript [30]. Battaglini et al. [30] analyzed, as a secondary endpoint, the role of neuromonitoring, including automated pupillometry, in predicting the occurrence of neurological complications and the outcome of critically ill COVID-19 patients. In one study, the primary objective was to define the syncope and presyncope incidence and clinical features in hospitalized subjects during the acute SARS-CoV-2 infection [44]. Finally, Koh et al. [39] evaluated the occurrence of neurological manifestations and, within them, autonomic symptoms in acute COVID-19 patients. 

#### 3.4.1. Characterization of Autonomic Involvement Associated with SARS-CoV-2

Among the studies that focused on the characterization of SARS-CoV-2-related AD, a great heterogeneity was found in the selection of the control group. Six studies compared COVID-19 patients to healthy volunteers [31,34,36,40,41,46], while three other studies considered as control group subjects affected by other infectious diseases [37], pneumonia [94], and/or respiratory failure from different etiology [47]. Oates et al. [44] compared COVID-19 patients with syncope and/or presyncope to those not presenting those symptoms. Two studies compared the autonomic parameters of the acute COVID-19 patients with those recorded three months after the disease [48] or during prior hospitalization for other causes in the same population [35]. Finally, the study conducted by Koh et al. [39] had no control group. For details, refer to Table 2.

##### Heart Rate Variability (HRV)

Concerning the HRV evaluation, contrasting results were found among studies. The standard deviation of the RR intervals (SDNN) was the HRV measure most frequently analyzed, being considered in six studies [31,34,36,37,40,94]. This parameter was similar between moderate COVID-19 patients and healthy volunteers in two studies [31,36] and between critically ill COVID-19 subjects and septic patients [37]. Kaliyaperumal et al. [36] found that the SARS-CoV-2-infected population presented a significantly higher parasympathetic overtone, defined as a SDNN > 60 and a root mean square of successive differences between normal heartbeats (RMSSD) > 40, than controls. Lonini et al. [40] measured HRV (summarized as SDNN), mean heart rate (HR), and mean respiratory rate (RR) in COVID-19 subjects before and after walking, finding a lower SDNN, a higher mean RR and HR at baseline, and a lower tolerance for physical exertion, demonstrated by a higher post-walking HR despite lower walking cadence in infected subjects than in healthy volunteers [40]. Moreover, SDNN differed between COVID-19 patients and people with pneumonia from different etiologies in an unspecified direction [94]. Analyzing the SDNN circadian pattern, Hirten et al. [34] found that the mean amplitude of SDNN circadian pattern was higher in COVID patients than in healthy volunteers. 

RMSSD was considered in five reports. This HRV measure was found to be higher in moderate COVID-19 patients than in healthy volunteers in one study [36], but the same parameter did not differ between similar populations in another report [31] and between ICU-admitted SARS-CoV-2-infected patients and subjects with sepsis from other causes [37]. Khalpey et al. [94] found undefined differences in RMSSD between COVID-19 patients and people with pneumonia from other causes.

The number of consecutive RR intervals that differed more than 50 ms (NN50) [31] and the proportion of NN50 (pNN50) [36] were similar in hospitalized patients with COVID-19 and healthy controls, but the second parameter differed between critically ill subjects infected with SARS-CoV-2 and those presenting sepsis from other causes, being higher in the septic group [37].

Finally, Junarta et al. [35] found an overall reduced HRV, including RMSSD, pNN50, and the standard deviation of successive differences in RR intervals (SDSD), during the acute phase of the infection compared to the same patients during previous hospitalization for other reasons.

No difference in the standard deviation of the averages of RR intervals (SDANN) was observed by Bellavia et al. [31] between moderate COVID-19 patients and healthy controls.

A geometric measure, the HRV triangular index, was analyzed in one study, and this parameter was found to be different between COVID-19 hospitalized patients and patients with pneumonia from different etiologies.

The frequency parameters of HRV were analyzed by four studies. Milovanovic et al. [41] found that the overall acute COVID-19 population presented a lower low frequency (LF) component than healthy volunteers, while the high frequency (HF) component was lower only in milder hospitalized patients, and the LF/HF was higher in those presenting a severe infection than in controls. The very low frequency (VLF) component did not differ among all the study groups. Considering two studies that compared moderate COVID-19 patients to healthy volunteers, the infected population presented lower HF and LF components than the controls in one study [36], while in the other report, frequency parameters did not differ between groups [31]. In the comparison between critically ill patients, people infected with SARS-CoV-2 had significantly lower HF, LF, and VLF components than patients with sepsis from other causes [37]. 

Concerning the non-linear measurements of HRV, two studies found that the two standard deviations of the mean RR interval (SD1, SD2) were similar between COVID-19 patients and the control group, composed of healthy volunteers [41] in one study and of septic patients in another [37]. The ratio of standard deviation derived from the Poincaré plot (SD1:SD2) did not differ between the overall SARS-CoV-2-infected population and healthy volunteers, but this ratio was significantly higher in critically ill patients with COVID-19 than in the septic group [37].

Finally, Kamaleswaran et al. [37] reported that acceleration capacity (AC) and the mode of the NN interval were significantly higher, while approximate entropy (ApEn), sample entropy (SamEn), and deceleration capacity (DC) were lower in COVID-19 subjects than in septic patients.

Skazkina et al. [46] compared the total percent of phase synchronization (S index) of the physiological oscillations of cardiovascular parameters, as measured by the HRV and the photoplethysmographic signals, between patients with acute SARS-CoV-2 infection and healthy volunteers. These authors found that the S index and, consequently, the degree of synchronization of the cardiovascular autonomic network, was significantly lower in the COVID-19 group than in controls.

##### Dynamic and Static Pupillometric Parameters

Concerning the automated pupillometry, two studies analyzed dynamic pupillometry [31,47], and one study investigated both dynamic and static pupillometric parameters [48]. In the static assessment, Yurttaser Ocak et al. [48] found that the enrolled subjects presented significantly lower scotopic and mesopic pupil diameters during the infection than 3 months later. The same authors described that, in the dynamic evaluation, the same population presented lower pupillary dilation velocities (DV) and mean pupil diameters at several timepoints after the light stimulus during hospitalization than at the subsequent evaluation [48]. On the other hand, moderate COVID-19 patients presented higher sympathetic (i.e., DV), parasympathetic (i.e., pupil constriction index (CH), absolute constriction amplitude (ACA)), and mixed (i.e., BPD) pupillary parameters than healthy subjects [31]. However, Vrettou et al. [47] did not find any difference in the dynamic automated pupillometry evaluation between critically ill COVID-19 patients and subjects with respiratory failure from other causes. 

##### Other Autonomic Parameters

Higher rate of feet sudomotor dysfunction, defined as a mean electrochemical skin conductance (ESC) < 70 μS, was found in moderate COVID-19 patients than in healthy controls through Sudoscan^®^ (Impeto Medical, Paris, France) [31]. The same authors did not find any differences in the pulse transit time (PTT) between the two groups [31].

The study of Milovanovic et al. [41] was the only one that analyzed the cardiovascular reflex tests (CART), the blood pressure variability (BPV), and the baroflex sensitivity (BS) in a group of moderate COVID-19 patients and in healthy volunteers. To summarize, in the CART examination, COVID-19 patients presented more frequently a combined AD and a sympathetic dysfunction than the controls, while the parasympathetic impairment was higher in mild and lower in severe patients than in healthy volunteers. Moreover, the SARS-CoV-2-infected group had higher systolic and diastolic HF and diastolic VLF components and lower diastolic LF and LF/HF components of BPV than the controls, as well as lower BRS.

Two studies reported symptoms compatible with an autonomic involvement in acute COVID-19 subjects through the retrospective evaluation of patients’ medical reports [39,44]. Oates et al. [44] found a prevalence of syncope/presyncope of 3.7% in hospitalized COVID-19 patients. Among these patients, an autonomic pathogenesis could be suspected in 9 out of 32 (28%) syncopal events based on the evidence of a neurocardiogenic involvement or a hypotensive mechanism. In particular, orthostatic hypotension was found in 50% of the hypotensive group. Koh et al. [39] found a prevalence of 0.01% of dysautonomic symptoms in the overall acute COVID-19 population of Singapore. Among these patients, three presented pupil abnormalities, two presented postural orthostatic tachycardia syndrome (POTS), and one patient presented small fiber neuropathy diagnosed by a sympathetic skin response test.

All three studies that compared symptomatic to asymptomatic COVID-19 patients found no differences in HRV parameters between the two groups [34,36,94].

#### 3.4.2. Effects of Autonomic Alterations on SARS-CoV-2 Infection Outcome

The selection of the primary outcome measures varied widely among reports, and death was the most frequent measure, since it was considered in four studies [29,35,37,43]. Other outcome measures were the increase in C-reactive protein (CRP) levels [32], ARDS development [42], COVID-19 severity [45], length of stay (LOS) of SARS-CoV-2-infected patients [38] and a composite outcome including ICU referral and ICU LOS, need for endotracheal intubation, and mortality [35]. 

Among time-domain HRV parameters, SDNN proved to be the strongest prognostic factor, confirming this role in four out of five studies (80%) [29,32,37,43,45]. To summarize, low SDNN appeared to be predictive of poor COVID-19 prognosis, being related to severe forms of the disease [45], ICU referral during hospitalization [43], and higher risk of short-term mortality, especially in older patients [43]. Moreover, a drop of ≥40% in SDNN value significantly predicted a ≥50% increase in CRP during the following 72 h [32]. Although in two reports [29,37] SDNN was not found to be a predictor of mortality in critically ill COVID-19 patients, Aragón-Benedí [29] found that low SDNN was associated with higher sequential organ failure assessment (SOFA) scores and lower time of survival in critically ill patients. 

RMSSD, measured in four studies [35,37,43,45], did not predict prognosis in terms of mortality [37,43], COVID-19 severity [45], hospital LOS, and need for mechanical ventilation [35]. Contrasting results were found concerning the need for ICU referral, since in the study conducted by Mol et al. [43], low RMSSD significantly predicted ICU admission during hospitalization, while in another report, this correlation was not found [35].

pNN50 was analyzed in two studies [35,37], confirming its prognostic role in predicting mortality but with opposite results; while Kamaleswaran et al. [37] reported higher pNN50 in non-survivors than in critically ill survivor patients, Junarta et al. [35] found that a reduced pNN50 value was related to an increased mortality and a lower rate of 60-day survival.

SDANN predicted COVID-19 severity in the only study that analyzed this parameter [45]. Finally, SDSD, considered in one study [35], did not predict prognosis in terms of mortality, hospital LOS, ICU referral, ICU LOS, and need for mechanical ventilation.

Two reports, analyzing the cardiac DC role as a prognostic factor of COVID-19 [37,42], found that a reduced DC was predictive of poor prognosis, both in terms of ARDS development [42] and death [37].

Concerning the frequency parameters of HRV, both HF and LF were not significantly different between critically ill COVID-19 survivors and non-survivors [37], and LF was not able to forecast the hospital LOS in another report [38]. On the other hand, Aragón-Benedí et al. [29] found that a HF higher than 80 n.u. predicted mortality with a sensitivity of 100% and a specificity of 80%. Finally, a reduction of LF/HF ratio during the SARS-CoV-2 infection was correlated with a shorter time to viral RNA negative conversion and recovery from disease in one study [45].

Kohdadadi et al. [38] considered, in addition to the aforementioned LF component of HRV, the respiratory sinus arrythmia amplitude (RSA), heart period (HP), and vagal efficiency (VE). A higher VE predicted a shorter LOS only in patients aged <40 years, while other parameters had no prognostic value.

Finally, Kamaleswaran et al. [37] reported that ApEn and SampEn were lower, while SD1:SD2, AC, and the mode of the NN interval were higher in COVID-19 non-survivors than in survivors.

Concerning the only study that analyzed the automated pupillometry role in predicting COVID-19 prognosis, analyzing the raw data of Battaglini et al. [30], although no difference was found in the dynamic pupillometric parameters between critically ill COVID-19 survivors and non-survivors, the pupillary constriction velocity (CV) was significantly lower in the COVID-19 patients who presented neurological complications. All the data of interest regarding this topic are summarized in Table 3.

#### 3.4.3. Role of Autonomic Parameters in Predicting SARS-CoV-2 Infection

Three studies analyzed the role of autonomic parameters as early markers of the infection, and all these reports employed HRV measured by wearable devices as the measure of choice [33,40,95]. 

The two reports including the participants of the Weltory study [96] found contrasting results; Ponomarev et al. [95] did not find differences in SDNN and RMSSD values before, during, and after SARS-CoV-2 infection, while Hijazi et al. [33] reported lower mean SDNN, RMSSD, pNN50 and higher LF/HF and HR during the infection than in the “healthy time”. Moreover, a model composed of five HRV variables (pNN50, RMSSD, LF/HF, SDNN, and HR) and the answer to the question “How do you feel?” was found to be able to discriminate the infection period from the healthy time [33]. It must be taken into account that while the second work considered all the 186 Weltory participants with available data on HRV parameters [33], the first one considered only 14 individuals who provided more than five high-quality measurements in the period before and during COVID-19 [95].

Lonini et al. [40] found that a model composed of mean HR and walking cadence only had a slightly lower predictive capacity (area under the curve—AUC = 0.908) in discriminating between SARS-CoV-2-infected patients and healthy volunteers compared to a model that analyzed several other variables, including the SDNN (AUC = 0.921).

A detailed representation of these data is available in Table 4.

## 4. Discussion

Despite limitations related to the inhomogeneous selection of the study population and the methodology for detecting the involvement of the ANS, this systematic review can conclude that acute SARS-CoV-2 infection leads to AD already during the acute stages of the disease in terms of cardiovascular, sudomotor, and pupillometric functions. Moreover, autonomic alterations occurring during the acute stage of COVID-19 are often associated with a worse outcome. Finally, the monitoring of autonomic parameters could lead to an early suspicion of the infection, even before the symptoms’ occurrence, but this hypothesis needs further validation.

Concerning the included studies, they were widely heterogeneous in terms of sample size, method of infection diagnosis or SARS-CoV-2 detection, COVID-19 severity, mean age of enrolled participants, tools employed to assess the autonomic functioning, and selection of a control group, making the summarization process extremely difficult. Moreover, only three studies were considered as “good” in the quality assessment [30,38,42].

Evaluating the characterization of autonomic involvement during the acute SARS-CoV-2 infection, most of the studies analyzed HRV to evaluate the ANS functioning [31,34,35,36,37,40,41,46,94]. Almost all studies certified an alteration of the HRV during the acute phase of SARS-CoV-2 infection [34,35,36,37,41,46,94], although the sympathetic or parasympathetic involvement was not univocal. In fact, probably due to a wide variety of recording tools (from wearable devices to continuous EKG monitoring), of the analyzed HRV parameters (i.e., time-domain, frequency-domain, and non-linear measures), of the recordings’ length (from a few seconds to over 24 h), it is difficult to interpret which tone between parasympathetic or sympathetic is prevalent. However, attempting to summarize, one study reported the occurrence of a parasympathetic predominance [36], another one a parasympathetic withdrawal [35], another study did not find differences in HRV parameters between non-critically ill COVID-19 patients and healthy controls [31], while two studies found a complex misalignment of ANS [37,41], involving both parasympathetic and sympathetic branches. Therefore, HRV parameters are strongly influenced by demographic factors, such gender and age [97], which varied widely among the included studies, and they could be altered by several clinical conditions [97,98,99], such as infectious diseases themselves [12,100,101,102]. 

The challenging question is whether HRV alterations are closely related to COVID-19-related dysautonomia or whether this involvement is a consequence of a generalized hyperinflammatory state, which could be associated with several other diseases. It is noteworthy that, although HRV differed between COVID patients and healthy controls in most of the included reports [34,36,40,41], a similar ASN involvement was found in symptomatic and asymptomatic COVID-19 patients [34,36,94], and significant differences in HRV were reported between patients affected by COVID-19 and pneumonia [94] or sepsis from other etiologies [37]. A further confirmation of the unique relationship between acute-phase COVID-19 and dysautonomia is provided by one study that analyzed the same subjects before and during the SARS-CoV-2 infection, confirming the alteration of HRV in the course of acute infection [35]. 

The other tools assessing autonomic parameters confirmed the involvement of ANS in the acute phase of COVID-19 [31,41,47,48]. Bellavia et al. [31], in the sudomotor and pupillometric analysis, found that non-critically ill hospitalized patients with acute COVID-19 presented both sympathetic and parasympathetic impairment. Similarly, Yurttaser Ocak et al. [48], in both static and dynamic pupillary evaluation, found a complex misalignment of ANS during the infection. Only one study found no differences in the automated pupillometry evaluation [47], probably due to the comparison between critically ill COVID-19 patients and ICU-admitted patients with respiratory failure from other causes, since pupillometric parameters are widely influenced by anesthetics, analgesics, and ionotropic medications usually employed in ICU [103]. Milovanovic et al. [41] found a complex misalignment of ANS, mostly characterized by a sympathetic withdrawal followed by a parasympathetic hyperactivity, in almost all the diagnostic tests employed, such as CART, BPV, HRV, and BRS. Moreover, the occurrence of verified “autonomic syndromes” in patients with acute COVID-19, such as orthostatic hypotension [44], POTS, hyperhidrosis, pupil abnormalities, and small fiber neuropathy [39], further supports the hypothesis of an autonomic involvement during the acute phase of the disease. In a recent published study, we confirmed a higher prevalence of orthostatic hypotension in acute COVID-19 patients than in healthy controls [104].

Based on the above findings, we can assume that acute SARS-CoV-2 infection leads to autonomic impairment, although which branch is predominantly involved between sympathetic and parasympathetic does not seem easily understandable, probably due to the complex imbalance between these systems.

Concerning the prognostic value of ANS involvement in determining the COVID-19 prognosis [29,30,32,35,37,38,42,43,45], obviously, the same observations as those mentioned above can be made regarding the lack of homogeneity of the records employed to analyze the HRV, both in terms of traces length and instruments used for measurements, and of the HRV parameters considered. However, all the studies that considered HRV as a prognostic factor found that at least one HRV parameter was able to predict COVID-19 prognosis, regardless of the HRV measure considered. SDNN was a stronger predictor of prognosis, since its reduction predicted an unfavorable prognosis of COVID patients in four out of five studies [29,32,37,43,45]. Concerning other HRV parameters, contrasting results were found regarding their prognostic role in acute COVID-19 patients, except for the DC, whose reduction was associated with a poor prognosis [37,42]. These results confirm the evidence of previous studies, which report that a reduction in SDNN and DC is associated with a poor prognosis in several medical conditions [105,106,107,108,109], including pneumonia from other causes [110] or sepsis [111]. 

On the other hand, in the only study in which another tool to detect dysautonomia was employed, no differences in automated pupillometry evaluation were found between ICU-admitted COVID-19 survivors and non-survivors [30]. However, the data regarding the prognostic role of pupillometric assessment were derived by the author of this review from the raw data, and this analysis was not the original aim of the study itself. 

With these assumptions, we can state that the prognostic role of AD during the acute SARS-CoV-2 infection is the strongest evidence emerging from this review, considering the large number of concordant studies. 

However, it is not clear whether ANS involvement worsens the prognosis of COVID-19 patients or whether the autonomic misalignment is just a marker of a severe disease leading to an intense proinflammatory state, a condition able to modify the autonomic balance [112,113,114]. Indeed, all the populations recruited in the studies analyzing this topic included hospitalized patients in regular or sub-intensive wards [35,38,42,43,45] or in ICU [29,30,32,37], suggesting a selection bias of patients with a more severe SARS-CoV-2 infection than the general COVID-19 population. Moreover, in several studies, the levels of proinflammatory markers correlated with HRV parameters [78,94] or clinical scores of disease severity [78], supporting the hypothesis that severe forms of the disease could lead to a dysregulation of the immune system and a consequent ANS imbalance. On the other hand, the ability of HRV parameters to predict not only the short-term but also the medium-term COVID-19 outcome [29,35,43,45] or intensive care unit admission [43], and the SDNN ability to foresee the rise in CRP over the next 72 h, [32] seem to support the hypothesis that AD precedes the clinical and laboratory worsening of the disease and COVID-19-related inflammation. Further controlled studies, including mild COVID-19 patients and using various tools to analyze the ANS functioning, should be performed in order to clarify this point.

Finally, in two out of three studies analyzing the role of autonomic parameters in predicting SARS-CoV-2 infection, HRV was found to have a good predicting capacity [33,40]. These findings could suggest that an autonomic imbalance could be an early feature of the disease, presenting even before the respiratory involvement. However, in one of these studies, the SARS-CoV-2 infection was self-reported by the study participants [33], and in the other, the test used to perform the COVID-19 diagnosis was not specified [40]. Moreover, both studies considered the PRV obtained from wearable devices as a surrogate of HRV [33,40]. Although the ability of HRV measured by wearable sensors to predict infection has already been verified for other viral diseases [115], no clear evidence on the role of AD in predicting COVID-19 could be affirmed from this review. Further studies, using various autonomic parameters and comparing the AD predictive capacity in identifying the SARS-CoV-2 infection compared to other infectious diseases, are needed in order to clarify this point.

The main limitation of this review is the low amount of good-quality studies included. As mentioned before, the included reports were greatly heterogeneous on several critical points. Moreover, the tool employed to evaluate the AD was HRV in most studies, and this parameter alone is not enough to define the presence of an autonomic imbalance and its direction. Finally, the protocol of this review was not registered in an online systematic review database (i.e., the international prospective register of systematic reviews - PROSPERO), so the extracted data could be altered by selection biases. Further studies, conducted on larger samples and analyzing the autonomic functioning from several points of view, are needed to reinforce the evidence emerging from this review, especially concerning the role of autonomic parameters in predicting the SARS-CoV-2 infection.

## 5. Conclusions

SARS-CoV-2 is associated with ANS involvement already during the early stage of the disease, which is associated with a poor prognosis. Further evidence is needed in order to clarify the role of ANS alterations in predicting the SARS-CoV-2 infection. The results emerging from this review should encourage future research to investigate the pathogenic mechanism of SARS-CoV-2-related ANS involvement and to identify patients with a high risk of developing AD and severe infection diseases. We can therefore hypothesize that a routinary examination of AD in COVID-19 patients could lead to a prognostic stratification of acute SARS-CoV-2 infection.

## Figures and Tables

**Figure 1 jcm-11-03883-f001:**
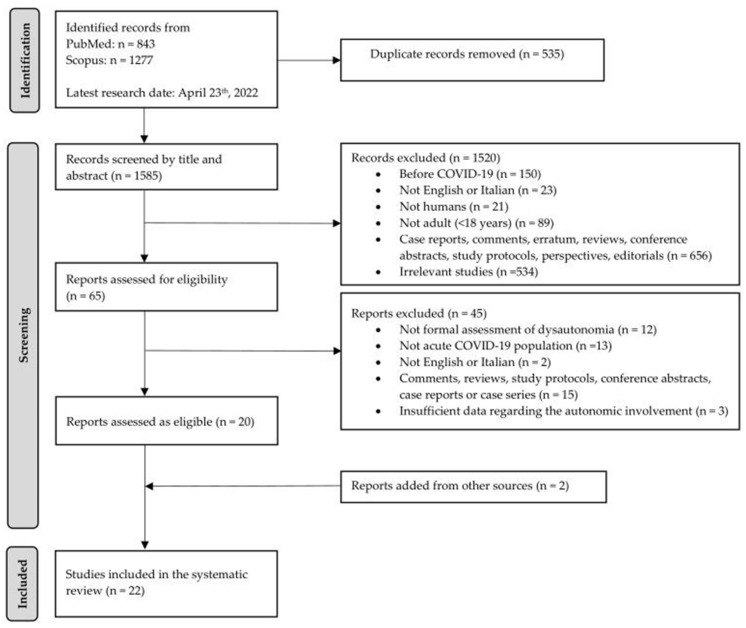
The PRISMA diagram of the systematic review. Reports excluded: Not formal assessment of dysautonomia—Refs. [66,67,68,69,70,71,72,73,74,75,76,77]; not acute COVID-19 population—Refs. [78,79,80,81,82,83,84,85,86,87,88,89,90]; not in English or Italian—Refs. [49,50]; study design—Refs. [51,52,53,54,55,56,57,58,59,60,61,62,63,64,65]; insufficient data regarding the autonomic involvement—Refs. [91,92,93]. Reports assessed as eligible—Refs. [29,30,31,32,33,34,35,36,37,38,39,40,41,42,43,44,45,46,47,48]. Reports added from other sources—Refs. [94,95]. Studies included in the systematic review—Refs. [29,30,31,32,33,34,35,36,37,38,39,40,41,42,43,44,45,46,47,48,94,95]. Abbreviations: COVID-19, Coronavirus disease 2019.

**Figure 2 jcm-11-03883-f002:**
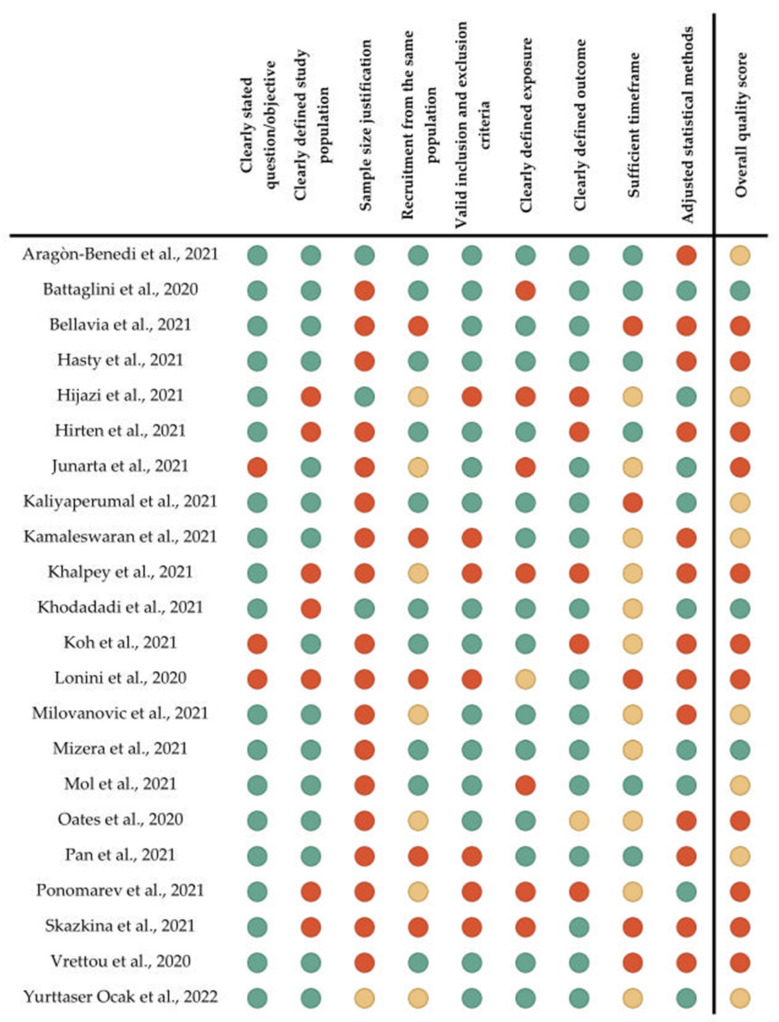
A summary of the quality assessment of the included studies according to the “Study Quality Assessment Tool” issued by the National Heart, Lung, and Blood Institute [29,30,31,32,33,34,35,36,37,38,39,40,41,42,43,44,45,46,47,48,94,95]. Color legend: Green, yes; Yellow, not applicable, not reported, or cannot be determined; Red, no.

**Table 1 jcm-11-03883-t001:** Inclusion and exclusion criteria of the systematic review according to PICOS.

	Inclusion Criteria	Exclusion Criteria
**Population**	Adult (≥18 years) patients with ongoing SARS-CoV-2 infection as diagnosed by a laboratory test, a thorax CT scan, self-reported by study participants, or reported by the authors of the study	Patients recovered from COVID-19, patients without history of SARS-CoV-2 infection, and patients with a dubious diagnosis of COVID-19
		Non-human sample
**Intervention**	NA	NA
**Comparison**	NA	NA
**Outcomes**	Characterization of autonomic involvement during acute SARS-CoV-2 infection	Absence of a formal evaluation of autonomic parameters
	Effects of autonomic alterations on the SARS-CoV-2 infection outcome	Lack of sufficient data regarding the autonomic assessment
	Role of autonomic parameters in predicting SARS-CoV-2 infection	
**Study design**	Case–control, cohort, cross-sectional, pre–post studies	Literature reviews, case reports or case series, conference papers, comments, editorials, erratum, study protocols, perspectives
	Published from 1 December 2020 to 23 April 2022	Not English or Italian Language

**Table 2 jcm-11-03883-t002:** A summary of the data extracted from the 13 studies, which characterized acute COVID-19-related autonomic dysfunction.

Authors, Year	Study Design	Study Population	Demographic Characteristics	COVID-19 Severity	COVID-19 Diagnosis	Dysautonomia Assessment	Study Endpoints	Main Findings
Bellavia et al., 2021 [31]	Observational, cross-sectional	20 acute COVID-19 patients vs. 20 healthy controls	COVID+ group: Mean age 56.1 ± 19.2 y; 70% men COVID- group: Mean age 52.6 ± 13.7 y; 65% men	Moderate	RT-PCR	Sudoscan, automated pupillometry (Npi-200), HRV measured from a 10 min long EKG in the lying position and a 3 min long EKG in the standing position, PTT measured with a pulse oximeter	Characterization of the autonomic nervous system involvement in acute COVID-19 patients	Pupillometry: COVID+ group presented higher CV, ACA, BPD, and CH than controls. Sudoscan: COVID+ patients presented feet sudomotor more frequently than controls. No differences between groups in terms of HRV parameters and PTT
Hirten et al., 2021 [34]	Prospective, observational, cohort	297 healthcare workers reporting data from wearable devices. 13/297 patients (COVID+)	Overall population: Mean age 36.3 ± 9.8 y; 31.6% men	NR	Self-reported RT-PCR test	PRV and HR measured by the PPG signal of wearable devices	Primary endpoint: Differentiation of acute COVID-19 patients from healthy controls through HRV. Secondary endpoints: Evaluation of HRV ability in predicting SARS-CoV-2 infection and in discriminating symptomatic and asymptomatic forms of COVID-19	Amplitude of SDNN lower in COVID+ than in COVID- groups and higher in uninfected participants than in COVID+ subjects during the 7 days prior and after a COVID-19 diagnosis. No HRV differences between symptomatic and asymptomatic COVID+ subjects
Junarta et al., 2021 [35]	Retrospective, observational, pre–post	38 hospitalized acute COVID-19 patients with chronic atrial fibrillation	Mean age 78.6 ± 11.4 y; 44.7% men	Moderate and severe	NR	HRV measured by EKGs obtained during hospitalization in the pre-COVID period and during admission for acute SARS-CoV-2 infection	Primary endpoint: Presence of HRV changes between the pre-COVID and the COVID period	HRV (SDSD, RMSSD, pNN50) significantly reduced during acute COVID-19
Kaliyaperumal et al., 2021 [36]	Observational, cross-sectional	63 acute COVID-19 patients vs. 43 age- and sex-matched healthy controls	COVID+: Mean age 48.4 ± 16.3 y; 69.8% men vs. COVID-: Mean age 50.1 ± 10.5 y; 62.8% men	Moderate	RT-PCR	HRV measured by a 5 min long EKG	Comparison of HRV parameters between acute COVID-19 patients and healthy controls	Lower values of HF, LF, and higher values of RMSSD in COVID-19 patients than in controls. Higher parasympathetic overtone (SDNN > 60 and/or RMSSD > 40) in COVID-19 patients than in healthy subjects. No HRV differences between symptomatic and asymptomatic COVID-19 patients
Kamaleswaran et al., 2021 [37]	Retrospective, observational, case–control	141 acute, ICU-admitted COVID-19 patients vs. 208 ICU-admitted patients with sepsis from other causes	COVID+: Mean age 63 ± 16 y; 52% men Septic patients: Mean age 63 ± 16 y; 55% men	Severe	RT-PCR	Average of HRV parameters measured by several 300 s sliding windows obtained from continuous bedside monitoring within 5 days of ICU admission	Secondary endpoint: Comparisons of HRV parameters between COVID-19 patients and patients with sepsis from other causes	COVID-19 patients presented lower median DC, ApEn, SampEn, pNN50, and higher median AC, SD1:SD2, and NN mode than sepsis patients
Khalpey et al., 2021 [94]	Retrospective, observational, case–control	200 patients divided into four groups: symptomatic COVID+, COVID+ with silent hypoxia, symptomatic COVID-, COVID- with silent hypoxia	NR	Not specified (moderate and/or severe)	RT-PCR	HRV measured by an EKG	Determination of HRV changes in patients with COVID-19 pneumonia	RMSSD, SDNN, and HRV triangular index differed between COVID+ and COVID- patients. The same parameters did not differ between symptomatic COVID+ patients and COVID+ subjects with silent hypoxia
Koh et al., 2021 [39]	Prospective, observational, cohort	47,572 acute COVID-19 patients	Overall population: Median age 34 y; age range 1−102 y; 98% men 5/47,572 patients with autonomic symptoms: Mean age 37.8 ± 6.6 y; 100% men	Mild, moderate, and severe	RT-PCR and/or raised IgG-anti SARS-CoV-2	Tilt table test, sympathetic skin response, and ophthalmological evaluation	Primary endpoint: Definition of neurological symptoms incidence and characterization in acute COVID-19 patients	Dysautonomic symptoms’ incidence in acute COVID-19 patients is 0.01%. Three patients presented pupil abnormalities (one—Adie’s pupil; one—Argyll Robertson; one—inverse Argyll Robertson), two patients presented POTS (one of them also presented Adie’s pupil and the other hyperhidrosis), and one patient presented small fiber neuropathy
Lonini et al., 2020 [40]	Observational, cross-sectional	15 acute COVID-19 patients vs. 14 healthy controls	Demographics are available for 14/15 COVID+ patients and 12/14 controls Overall COVID-19 population: Mean age 52.0 ± 15.2 y; 50% men Healthy controls: Mean age 32.4 ± 6.8 y; 67% men	Mild and moderate	The diagnostic test employed not specified	PRV measured by wearable devices and a sensing platform during periods of rest, walking, and forced coughs	Ability of physiological parameters measured by wearable sensors and sensing platforms in the discrimination between COVID-19 patients and healthy controls	SDNN at rest significantly lower in COVID-19 patients than in controls at baseline (pre-walk). No change in SDNN of COVID-19 patients during and after exercise. Higher post-exercise heart rate in COVID-19 patients than in healthy controls, despite the lower walking cadence
Milovanovic et al., 2021 [41]	Retrospective, observational, case–control	75 acute COVID-19 patients Mild group 30/75 (no pneumonia) Severe group 45/75 (with interstitial pneumonia) vs. 77 age-matched healthy controls	Mild COVID-19 patients: Mean age 41.6 ± 16.7 y; 53% men Severe COVID-19 patients: Mean age 51.3 ± 19.1 y; 53% men Sex- and age-matched controls	Moderate	RT-PCR	CART (Valsalva ratio, deep breathing test, blood pressure response to standing, handgrip test), HRV measured by continuous bedside monitoring, BPV and BRS measured by continuous bedside monitoring	Evaluation of autonomic dysfunction in acute COVID-19 patients and its impact on the cardiovascular system	CART: COVID-19 population presented higher prevalence of combined autonomic dysfunction and sympathetic dysfunction than controls. Parasympathetic dysfunction more frequent in mild cases and lower in severe cases of COVID-19 than in the control group. HRV: LF significantly lower in COVID-19 patients than in the control group. HF significantly lower in mild and LF/HF in severe COVID-19 patients than in controls. SD1 and SD1:SD2 lower in mild COVID-19 patients than in controls. BPV: higher systolic and diastolic HF and diastolic VLF and lower diastolic LF and LF/HF in COVID-19 patients than in controls. BRS: lower BRS in COVID-19 patients than in controls
Oates et al., 2020 [44]	Retrospective, observational, case–control	37 acute COVID-19 patients with syncope/presyncope vs. 40 acute COVID-19 patients without syncope/presyncope	Overall population: Median age 69 (56–73) y; 55% men Syncope group: Median age 69 (56.5–73) y; 51% men No syncope group: Median age 68 (56–73) y; 57% men	Moderate and severe	RT-PCR	Evaluation of patients’ medical records, including heart rate and blood pressure measurements	Definition of syncope/presyncope incidence, characteristics, and outcomes in acute COVID-19 patients	Syncope/presyncope incidence was 3.7% (37/1000). Syncope/presyncope patients hospitalized less frequently in ICU settings. Within syncope subtypes, 12.5% (4/32) were hypotensive, and 15.6% (5/32) were neurocardiogenic. Two out of four patients with hypotensive syncope presented orthostatic hypotension, while the other two were not tested
Skazkina et al., 2021 [46]	Observational, cross-sectional	32 acute COVID-19 patients vs. 32 healthy controls	COVID+ group: Age range 25–68 y; 56.3% men COVID- group: Age range 17–23 y; 31.3% men	NR	NR	HRV measured by 20 min long EKG and PPG	Comparison of the degree of synchronization of the autonomic control loops of circulation between COVID-19 patients and controls	Mean phase synchronization index was lower in acute COVID-19 patients than in healthy controls
Vrettou et al., 2020 [47]	Observational, cross-sectional	41 patients requiring mechanical ventilation for at least 48 h COVID+ group: 18/41 Respiratory failure from other etiologies: 23/41	COVID+ group: sedated patients (6/18) median age 68 (55–76) y; 83% men vs. not-sedated patients (12/18) median age 68 (60–78) y; 75% men COVID- group: sedated patients (14/23) median age 65 (51–76) y; 71% men vs. not-sedated patients (9/23) median age 65 (56–80) y; 67% men	Severe	RT-PCR	Automated pupillometry (Npi-200)	Differences in pupillary reactivity between mechanically ventilated ICU patients with COVID-19 and patients with respiratory failure from other etiologies	No significant differences in pupillary reactivity between COVID+ and COVID- groups. BPD, CH, CV, MCV, and DV were higher in awake COVID-19 patients than in sedated COVID-19 group
Yurttaser Ocak et al., 2022 [48]	Prospective, observational, pre–post	58 acute COVID-19 patients	Mean age 47.23 ± 1.1 years; 56.9% men	Moderate	RT-PCR	Automated pupillometry (Sirius Topographer, Phoenix v2.1 software)	Comparison of pupillary reactivity during and three months after COVID-19 infection	Mean mesopic and scotopic diameter significantly lower during the acute phase of the infection than three months later. Mean pupil diameter and average DV significantly lower at different timepoints after the luminous stimulus during the acute phase of infection than after three months

Abbreviations: COVID-19, Coronavirus Disease 2019; y, years; RT-PCR, Real-Time Polymerase Chain Reaction; HRV, Heart Rate Variability; EKG, Electrocardiogram; PTT, Pulse Transit Time; CV, Constriction Velocity; ACA, Absolute Constriction Amplitude; BPD, Baseline Pupil Diameter; CH, Constriction Index; NR, Not Reported; PRV, Pulse Rate Variability; HR, Heart Rate; PPG, Photoplethysmography; SARS-CoV-2, Severe Acute Respiratory Syndrome Coronavirus 2; SDNN, Standard Deviation of the RR intervals; SDSD, Standard Deviation of Successive Differences in RR intervals; RMSSD, Root Mean Square of Successive Differences between normal heartbeats; pNN50, proportion of consecutive RR intervals that differs more than 50 ms; HF, High Frequency; LF, Low Frequency; ICU, intensive Care Unit; DC, Deceleration Capacity; ApEN, Approximate Entropy; SampEn, Sample Entropy; AC, Acceleration Capacity; SD1:SD2, ratio of standard deviation derived from the Poincaré plot; NN, RR interval; POTS, Postural Orthostatic Tachycardia Syndrome; CART, Cardiovascular reflex tests; BPV, Blood Pressure Variability; BRS, Baroreceptor Reflex Sensitivity; SD1, Standard Deviation of the RR interval; VLF, Very Low Frequency; MCV, maximum constriction velocity; DV, Dilation Velocity.

**Table 3 jcm-11-03883-t003:** A table summarizing the data extracted from the nine studies investigating the prognostic role of autonomic dysfunction in acute COVID-19.

Authors, Year	Study Design	Study Population	Demographic Characteristics	COVID-19 Severity	COVID-19 Diagnosis	Dysautonomia Assessment	Study Endpoints	Main Findings
Aragon-Benedì et al., 2021 [29]	Prospective, observational, cohort	14 acute ICU-admitted COVID-19 patients Survivors group: 7 Non-survivors group: 7	Survivors group: median age 64 (60–73) y; 57% men Non-survivors group: median age 71 (57–72) y; 100% men	Severe	RT-PCR	HRV measured by means of analgesia nociception index monitor from 240 s long EKG	Demonstration of an autonomic involvement with a sympathetic predominance and a parasympathetic withdrawal in the most severely ill COVID-19 patients	ANIm and ANIi (indices of the HF component) significantly higher in non-survivors than in survivors. Lower energy (SDNN) correlated with higher SOFA score and, in non-survivors, with fewer survival days. A limit value of 80 for ANIm predicted mortality (sensitivity of 100%; specificity of 85.7%). A limit value of 0.41 ms for energy predicted mortality (sensitivity of 71.4%; specificity of 71.4%)
Battaglini et al., 2020 [30]	Retrospective, observational, cohort	94 acute COVID-19 patients undergoing invasive mechanical ventilation 53/94 (56%) underwent continuous neuromonitoring In 29/94 (31%), pupillary reactivity was tested	Overall population: Mean age 61.6 ± 11.1 y; 78.7% men Patients with neurological complications: Mean age 62.4 ± 8.3 y; 87.2% men Patients without neurological complications: Mean age 60.8 ± 13.3 y; 70.2% men	Severe	RT-PCR	Automated pupillometry (Neurolight Algiscan)	Primary endpoint: Prevalence of neurological complications and their effects on outcome. Secondary endpoint: Role of cerebral hemodynamics changes in predicting outcome and occurrence of neurological complications	Neurological complications incidence was 50%, and they were associated with longer overall and ICU stay. Automated pupillometry evaluation did not discriminate outcome in terms of mortality. Patients with neurological complications presented lower mean CV than patients without neurological symptoms
Hasty et al., 2021 [32]	Prospective, observational, cohort study	16 acute COVID-19 patients requiring HFNO or mechanical ventilation	Mean age 60.5 ± 13.4 y; 71% men	Severe	RT-PCR	HRV measured by 7 min long single-limb EKG traces	Evaluation of a correlation between HRV reduction and CRP increment	A >40% decrease in SDNN predicted a subsequent 50% rise in CRP, with 83.3% sensitivity and 75% specificity
Junarta et al., 2021 [35]	Retrospective, observational, pre–post	38 hospitalized acute COVID-19 patients with chronic atrial fibrillation	Mean age 78.6 ± 11.4 y; 44.7% men	Moderate and severe	NR	HRV measured by EKGs obtained during hospitalization in the pre-COVID period and during admission for acute SARS-CoV-2 infection	Secondary endpoint: Prognostic role of HRV in acute COVID-19 patients, comparing patients with reduced HRV to ones with preserved HRV	Patients with reduced HRV presented higher mortality when stratified for pNN50
Kamaleswaran et al., 2021 [37]	Retrospective, observational, case–control	141 acute, ICU-admitted COVID-19 patients vs. 208 ICU-admitted patients with sepsis from other causes	COVID+: Mean age 63 ± 16 y; 52% men (survivors: Mean age 59 ± 15 y; 53% men. Non-survivors: Mean age 71 ± 14 y; 52% men)	Severe	RT-PCR	Average of HRV parameters measured by several 300 s sliding windows obtained from continuous bedside monitoring within 5 days of ICU admission	Primary endpoint: Analysis of HRV differences between survivors and non-survivors with acute COVID-19	SD1:SD2, AC, and pNN50 were higher, and NN, ApEn, SampEN, and DC were lower in COVID-19 non-survivors than in COVID-19 survivors
Khodadadi et al., 2021 [38]	Prospective, observational, cohort	36 acute COVID-19 patients	NR	Moderate and severe	RT-PCR	LF–HRV, RSA amplitude, heart period, and vagal efficiency measured by a 7–10 min long EKG obtained by means of a Polar H10 heart rate sensor on the first day of admission	Primary endpoint: Definition of the role of demographic, clinical, and HRV parameters in forecasting LOS of COVID-19 patients	A higher vagal efficiency correlated with shorter LOS only in younger patients (<40 y)
Mizera et al., 2021 [42]	Prospective, observational, cohort	60 acute COVID-19 patients with sinus rhythm ARDS group (37/60) vs. non-ARDS group (23/60)	Overall population: Mean age 66.9 ± 13.4 y; 60% men ARDS group: Mean age 69.1 ± 12.8 y; 56.8% men Non-ARDS group: Mean age 63.3 ± 13.9 y; 65.2% men	Moderate and/or severe	RT-PCR	DC measured by 10–30 min long EKG obtained from a 24 h EKG Holter recording	Evaluation of DC’s role in the prediction of ARDS development in acute COVID-19 patients	DC significantly lower in the ARDS group when compared to the non-ARDS group. Patients with ARDS were more likely to show DC < 4.5 ms. Decreased DC was also associated with increased risk of ARDS in the adjusted analysis and presented a good discriminatory capacity for predicting COVID-19 patients at risk of developing ARDS
Mol et al., 2021 [43]	Retrospective, observational, cohort	271 hospitalized, acute COVID-19 patients	Overall population: Mean age 68 y (age range 25 to 95 y); 59% men	Moderate and severe	RT-PCR	HRV measured by a 10 s long EKG performed within 3 days of admission	Primary endpoint: HRV ability to predict overall survival at three weeks from admission. Secondary endpoints: HRV ability to predict ICU referral and impact of HRV on other prognostic factors, such as age	Patients with SDNN > 8 had a lower risk of death than those with SDNN ≤ 8 at three weeks from admission. SDNN ≤ 8 predicted a higher mortality only in older patients (≥70 y). Only patients aged ≥70 y with low HRV were at higher risk of death than younger patients. Lower risk of needing ICU care in patients with SDNN > 8 and RMSSD > 8
Pan et al., 2021 [45]	Prospective, observational, cohort	34 acute COVID-19 patients Mild group: 13/34 Severe group: 21/34	Overall population: Mean age 56.2 ± 16.0 y; 32% men Mild group: Mean age 47.5 ± 14.2 y; 23% men Severe group: 61.5 ± 15.0 y; 38% men	Moderate and severe	RT-PCR	HRV measured by a 24 h long Holter EKG	Primary endpoint: Evaluation of HRV ability in prediction of severity and prognosis of acute COVID-19 patients	Lower SDNN, SDANN, and higher LF/HF in severe than in mild patients, and these parameters discriminated the two conditions with a good sensitivity and specificity. Shorter time to viral RNA negative conversion and disease recovery in severe patients with decreased LF/HF than those with increased LF/HF

Abbreviations: ICU, Intensive Care Unit; COVID-19, Coronavirus Disease 2019; y, years; RT-PCR, Real-Time Polymerase Chain Reaction; HRV, Heart Rate Variability; EKG, Electrocardiogram; ANIm, mean Analgesia Nociception Index; ANIi, instantaneous Analgesia Nociception index; HF, High Frequency; SDNN, Standard Deviation of the RR intervals; SOFA, Sequential Organ Failure Assessment; CV, Constriction Velocity; HFNO, High-Flow nasal Oxygen; CRP, C-reactive protein; NR, Not reported; SARS-CoV-2, Severe Acute Respiratory Syndrome Coronavirus 2; pNN50, proportion of consecutive RR intervals that differs more than 50 ms; SD1:SD2, ratio of standard deviation derived from the Poincaré plot; AC, Acceleration Capacity; NN, RR interval; ApEN, Approximate Entropy; SampEn, Sample Entropy; DC, Deceleration Capacity; LF, Low Frequency; RSA, Respiratory Sinus Arrythmia; LOS, Length Of Stay; ARDS, Acute Respiratory Distress Syndrome; RMSSD, Root Mean Square of Successive Differences between normal heartbeats; SDANN, Standard Deviation of the Averages of RR intervals.

**Table 4 jcm-11-03883-t004:** A graphic representation of the data extracted from the three studies analyzing the autonomic parameters’ role in predicting the SARS-CoV-2 infection.

Authors, Year	Study Design	Study Population	Demographic Characteristics	COVID-19 Severity	COVID-19 Diagnosis	Dysautonomia Assessment	Study Endpoints	Main Findings
Hijazi et al., 2021 [33]	Retrospective, observational, pre–post	186 acute COVID-19 patients (Weltory study)	Mean age 44 ± 14.1 y; 36% men	NR	Self-reported by study participants	PRV and HR measured by the PPG signal of wearable devices	Evaluation of wearable devices’ role in predicting SARS-CoV-2 infection	SDNN, RMSSD, pNN50 significantly lower, and HR and LF/HF significantly higher during the infection than during the healthy time. The model with five HRV variables (pNN50, RMSSD, LF/HF, SDNN, and HR) and the answer to the question “How do you feel?” ranked the best in the discrimination between infection and healthy time, with an AUC value of 0.938
Lonini et al., 2020 [40]	Observational, cross-sectional	15 acute COVID-19 patients vs. 14 healthy controls	Demographics are available for 14/15 COVID+ patients and 12/14 controls Overall COVID-19 population: Mean age 52.0 ± 15.2 y; 50% men Healthy controls: Mean age 32.4 ± 6.8 y; 67% men	Mild and moderate	The diagnostic test employed not specified	PRV measured by wearable devices and a sensing platform during periods of rest, walking, and forced coughs	Evaluation of the ability of physiological parameters measured by wearable sensors and sensing platforms in the discrimination between COVID-19 patients and healthy controls	SDNN at rest significantly lower in COVID-19 patients than in controls at baseline (pre-walk). No change in SDNN of COVID-19 patients during and after exercise
Ponomarev et al., 2021 [95]	Retrospective, observational, pre–post	14 acute COVID-19 patients (Weltory study)	Mean age: 44 ± 8.7 y; 64% men	NR	Self-reported by study participants	PRV measured by the PPG signal of wearable devices at the time of day each user took measurements most often	Evaluation of HRV ability in the prediction of COVID-19 infection	No significant differences in HRV parameters before, during, and after acute COVID-19 in the overall population. Analyzing individual users independently, three users presented lower SDNN, one user higher SDNN, and one user lower RMSSD during COVID-19 compared to the period before the infection

Abbreviations: COVID-19, Coronavirus Disease 2019; y, years; NR, Not Reported; PRV, Pulse Rate Variability; HR Heart Rate; PPG, Photoplethysmography; SARS-CoV-2, Severe Acute Respiratory Syndrome Coronavirus 2; SDNN, Standard Deviation of the RR intervals; RMSSD, Root Mean Square of Successive Differences between normal heartbeats; pNN50, proportion of consecutive RR intervals that differs more than 50 ms; LF, Low Frequency; HF, High Frequency; HRV, Heart Rate Variability; AUC, Area Under the Curve.

## Data Availability

The protocol and all the data of this systematic review are available on request from the corresponding author.

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
