# Peer review of "Autonomic Dysfunction during Acute SARS-CoV-2 Infection: A Systematic Review"

_jcm, 2022, doi:10.3390/jcm11133883_

Round 1
Reviewer 1 Report
Starting with Table 2, the pages are not numbered correctly. As the authors note (Figure 2), the general quality of most of the studies they review is poor. Table 2, summarizing the findings, is difficult to read and shows the sample sizes for many of the studies are small and some of the COVID-19 diagnoses are based on self-report. I assume the primary purpose of the paper is to present evidence indicating that more studies of the role autonomic dysfunction (AD) during COVID-19 infection need to be conducted. The study addresses that issue but Table 2 needs to be simplified and perhaps the small and poorly designed studies and those not adjusted for confounding need to be eliminated. The authors may want to present separate tables for AD during COVID-19 infection and AD and survival.
Author Response
We thank the reviewer for the suggestions.
1) “Starting with Table 2, the pages are not numbered correctly.”
We apologize for the mistake. We have corrected the page numbering, as suggested.
2) “As the authors note (Figure 2), the general quality of most of the studies they review is poor. Table 2, summarizing the findings, is difficult to read and shows the sample sizes for many of the studies are small and some of the COVID-19 diagnoses are based on self-report. I assume the primary purpose of the paper is to present evidence indicating that more studies of the role autonomic dysfunction (AD) during COVID-19 infection need to be conducted. The study addresses that issue but Table 2 needs to be simplified and perhaps the small and poorly designed studies and those not adjusted for confounding need to be eliminated. The authors may want to present separate tables for AD during COVID-19 infection and AD and survival.”
We decided to include in the systematic review all the studies that met the inclusion criteria without presenting any exclusion criterion, regardless their limitations, according to the latest PRISMA guidelines. The reason of this choice is to be as inclusive as possible, in order to collect any available evidence. On the other hand, the goal of quality assessment is precisely to provide a tool to the reader for the critical interpretation of the included studies.
As suggested, we changed Table 2 as follows:
- We reported the included studies in three different tables (Table 2, Table 3, and Table 4) based on the outcome of the studies themselves. In particular, we reported in Table 2 all the studies that focused on the characterization of COVID-19 related AD, in Table 3 all the studies that investigated the prognostic role of AD, and in Table 4 the studies analyzing the role of autonomic parameters in predicting Sars-CoV-2 infection.
- We tried to simplify the results reported inside the tables as much as possible.

Reviewer 2 Report
The authors presented a really interesting systematic review on the problem of autonomic dysfunction in covid-19. I have just a little technical commentary:
1) add reference numbers to Table 2 for the reader's convenience.
2) explain why conference papers were not included in the review, even though many of them have quite detailed descriptions of methods and original results.
Author Response
We thank the reviewer for the suggestions.
1) “add reference numbers to Table 2 for the reader's convenience.”
We added the reference numbers to Table 2, as suggested. Furthermore, we decided to add the reference numbers also to Figure 2, the one reporting the results of the quality assessment.
2) “explain why conference papers were not included in the review, even though many of them have quite detailed descriptions of methods and original results.”
We decided not to include conference papers because these types of manuscripts report unpublished data that have not undergone the peer review process. Consequently, conference papers have never been subjected to a quality check.

Round 2
Reviewer 1 Report
Page 8, Table 2, add mean age to COVID+ in first article in table.
This version is better.
Author Response
We thank the reviewer for the suggestions.
- “Page 8, Table 2, add mean age to COVID+ in first article in table.”
We have inserted “mean age” in Table 2, as suggested by the reviewer.
